# ONE STEP DIFFUSION VIA SHORTCUT MODELS

**Kevin Frans**
UC Berkeley
kvfrans@berkeley.edu

**Danijar Hafner**
UC Berkeley

**Sergey Levine**
UC Berkeley

**Pieter Abbeel**
UC Berkeley

## ABSTRACT

Diffusion models and flow-matching models have enabled generating diverse and realistic images by learning to transfer noise to data. However, sampling from these models involves iterative denoising over many neural network passes, making generation slow and expensive. Previous approaches for speeding up sampling require complex training regimes, such as multiple training phases, multiple networks, or fragile scheduling. We introduce shortcut models, a family of generative models that use a single network and training phase to produce high-quality samples in a single or multiple sampling steps. Shortcut models condition the network not only on the current noise level but also on the desired step size, allowing the model to skip ahead in the generation process. Across a wide range of sampling step budgets, shortcut models consistently produce higher quality samples than previous approaches, such as consistency models and reflow. Compared to distillation, shortcut models reduce complexity to a single network and training phase and additionally allow varying step budgets at inference time.

## 1 INTRODUCTION

Iterative denoising methods such as diffusion (Sohl-Dickstein et al., 2015; Ho et al., 2020; Song et al., 2020) and flow-matching (Lipman et al., 2022; Liu et al., 2022) have seen remarkable success in modelling diverse images (Rombach et al., 2022; Esser et al., 2024), video (Ho et al., 2022; Bar-Tal et al., 2024), audio (Kong et al., 2020), and proteins (Abramson et al., 2024). Yet, their weakness lies in expensive inference. Despite producing high-quality samples, these methods require an iterative inference procedure—often requiring dozens to hundreds of forward passes of the neural network—making generation slow and expensive. We posit that there exists a generative modelling objective which retains the benefits of diffusion training, yet can denoise in a *single step*.

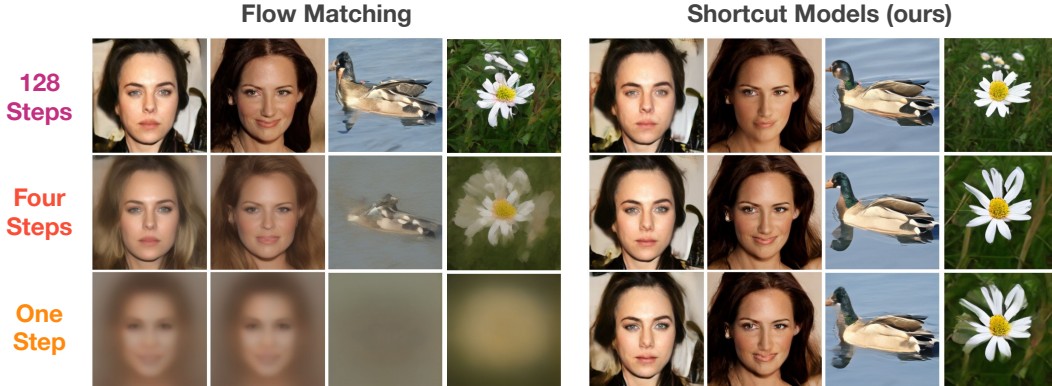

Figure 1: **Generations of flow-matching models and shortcut models for different inference budgets.** Shortcut models generate high-quality images across a wide range of inference budgets, including using a single forward pass, drastically reducing sampling time by up to 128x compared to diffusion and flow-matching models. In contrast, diffusion and flow-matching models rapidly deteriorate when queried in the few-step setting. The same starting noise used within each column and two models are trained on CelebA-HQ and Imagenet-256 (class conditioned).

We consider the *end-to-end* setting, in which one-step denoising is acquired by a single model over a single training run. Closely related are previous *two-stage* methods which take existing diffusion models and later distill one-step capabilities into them. These stages introduce complexity and require either generating a large synthetic dataset (Luhman & Luhman, 2021; Liu et al., 2022) or propagating through a series of teacher and student networks (Ho et al., 2020; Meng et al., 2023). Consistency models (Song et al., 2023) are step closer to the end-to-end setting, but their dependency on large amounts of bootstrapping requires a careful learning schedule throughout training. Two-stage or tightly-scheduled procedures suffer from a need to specify when to end training and begin distillation. In contrast, end-to-end methods can be trained indefinitely to continually improve.

We present shortcut models, a class of end-to-end generative models that produce high-quality generations under *any* inference budget, including in a single sampling step. Our key insight is to condition the neural network not only on the noise level but also the desired step size, enabling it to accurately jump ahead in the denoising process. Shortcut models can be seen as performing self-distillation *during training time*, and thus do not require a separate distillation step and are trained over a single run. No schedules or careful warmups are necessary. Shortcut models are efficient to train, requiring only $\sim 16\%$ more compute than that of a base diffusion model.

Empirical evaluations display that shortcut models satisfy a number of useful desiderata. On the commonly used CelebA-HQ and Imagenet-256 benchmarks, a single shortcut model can handle many-step, few-step, and one-step generation. Accuracy is not sacrificed —- in fact, many-step generation quality matches those of baseline diffusion models. At the same time, shortcut models can consistently match or outperform two-stage distillation methods in the few- and one-step settings.

The key contributions of this paper are summarized as follows:

- We introduce *shortcut models*, a class of generative models that generate high-quality samples in a single forward pass, by conditioning the model on the desired step size. Unlike distillation or consistency models, shortcut models are trained in a single training run without a schedule.
- We perform a comprehensive comparison of shortcut models to previous diffusion and flow-matching approaches on CelebAHQ-256 and ImageNet-256 under fixed architecture and compute. Shortcut models match or exceed the distillation methods that require multiple training phases and significantly outperform previous end-to-end methods across inference budgets.
- To demonstrate the generality of shortcut models beyond image generation, we apply them to robotic control and replace diffusion policies with shortcut policies. We observe that shortcut models maintain comparable performance under an order-of-magnitude lower inference cost.
- We release model checkpoints and the full training code for replicating our experimental results: https://github.com/kvfrans/shortcut-models

## 2 BACKGROUND

**Diffusion and flow-matching.** A recent family of models, including diffusion (Sohl-Dickstein et al., 2015; Ho et al., 2020; Song et al., 2020) and flow-matching[1] (Lipman et al., 2022; Liu et al., 2022) models, approach the generative modelling problem by learning an ordinary differential equation (ODE) that transforms noise into data. In this work, we adopt the optimal transport flow-matching objective (Liu et al., 2022) for simplicity. We define $x_t$ as a linear interpolation between a data point $x_1 \sim \mathcal{D}$ and a noise point $x_0 \sim \mathcal{N}(0, \mathbb{I})$ of the same dimensionality. The velocity $v_t$ is the direction from the noise to the data point:

$$x_t = (1-t)\,x_0 + t\,x_1 \qquad \text{and} \qquad v_t = x_1 - x_0. \tag{1}$$

Given $x_0$ and $x_1$, the velocity $v_t$ is fully determined. But given only $x_t$, there are multiple plausible pairs $(x_0, x_1)$ and thus different values the velocity can take on, rendering $v_t$ a random variable. Flow models learn a neural network to estimate the expected value $\bar{v}_t = \mathbb{E}[v_t \mid x_t]$ that averages over all plausible velocities at $x_t$. The flow model can be optimized by regressing the empirical velocity of randomly sampled pairings of noise $x_0$ and data $x_1$ pairs:

$$\bar{v}_\theta(x_t, t) \approx \mathbb{E}_{x_0, x_1 \sim D}\left[v_t \mid x_t\right] \qquad \mathcal{L}^{\mathrm{F}}(\theta) = \mathbb{E}_{x_0, x_1 \sim D}\left[||\bar{v}_\theta(x_t, t) - (x_1 - x_0)||^2\right] \tag{2}$$

---

[1]We consider flow-matching as a special case of diffusion modelling (Kingma & Gao, 2024), and use the terms interchangeably.

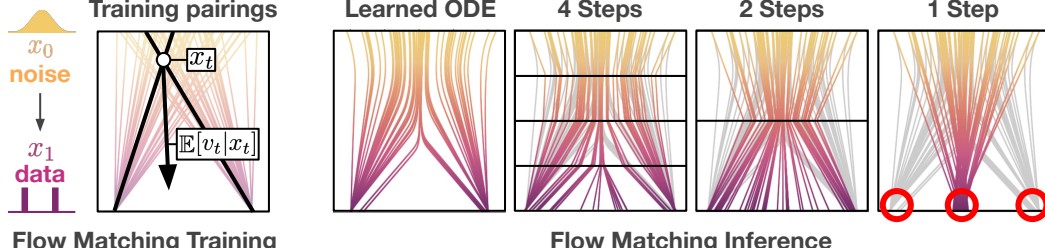

Figure 2: **Naive diffusion and flow-matching models fail at few-step generation. Left:** Training paths are created by randomly pairing data and noise. Note that the paths overlap; there is inherent uncertainty about the direction $v_t$ to the data point, given only $x_t$. **Right:** While flow-matching models learn a deterministic ODE, its paths are not straight and have to be followed closely. The predicted directions $v_t$ point towards the average of plausible data points. The fewer inference steps, the more the generations are biased towards the dataset mean, causing them to go off track. At the first sampling step, the model points towards the dataset mean and thus cannot generate multi-modal data in a single step **(see red circles)**.

To sample from a flow model, a noise point $x_0$ is first sampled from the normal distribution. This point is then iteratively updated from $x_0$ to $x_1$ following the the the *denoising ODE* defined as following the learned flow model $\bar{v}_\theta(x_t, t)$. In practice, this process is approximated using Euler sampling over small discrete time intervals.

**Few-step ambiguity.** While a perfectly trained ODE deterministically maps the noise distribution to the data distribution in continuous time, this guarantee is lost under finite step sizes. As illustrated in Figure 2, flow-matching learns to predict the *average* direction from $x_t$ towards the data, so following the prediction with a large step size will jump to an average of multiple data points. At $t = 0$ the model receives pure noise as input and $(x_0, x_1)$ are randomly paired during training, so the predicted velocity at $t = 0$ points towards the dataset mean. Thus, even at the *optimum* of the flow matching objective, one step generation will fail for any multi-modal data distribution.

## 3    SHORTCUT MODELS FOR FEW STEP GENERATION

We introduce shortcut models, a new family of denoising generative models that overcomes the large number of sampling steps required by diffusion and flow-matching models. Our key intuition is that we can train a single model that supports different sampling budgets, by conditioning the model not only on the timestep $t$ but also on a desired step size $d$.

As shown in Figure 2, flow-matching learns an ODE that maps noise to data along curved paths. Naively taking large sampling steps leads to large discretization error and in the single-step case, to catastrophic failure. Conditioning on $d$ allows shortcut models to account for the future curvature, and jump to the correct next point rather than going off track. We refer to the *normalized direction* from $x_t$ towards the correct next point $x'_{t+d}$ as the shortcut $s(x_t, t, d)$:

$$x'_{t+d} = x_t + s(x_t, t, d)\, d. \tag{3}$$

Our aim is to train a shortcut model $s_\theta(x_t, t, d)$ to learn the shortcut for all combinations of $x_t$, $t$, and $d$. Shortcut models can thus be seen as a generalization of flow-matching models to larger step sizes: whereas flow-matching models only learn the instantaneous velocity, shortcut models additionally learn to make larger jumps. At $d \to 0$, the shortcut is equivalent to the flow.

A naive way to compute targets for training $s_\theta(x_t, t, d)$ would be to fully simulate the ODE forward with a small enough step size (Luhman & Luhman, 2021; Liu et al., 2022). However, this approach is computationally expensive, especially for end-to-end training. Instead, we leverage an inherent self-consistency property of shortcut models, namely that one shortcut step equals two consecutive shortcut steps of half the size:

$$s(x_t, t, 2d) = s(x_t, t, d)/2 + s(x'_{t+d}, t + d, d)/2 \tag{4}$$

This allows us to train shortcut models using self-consistency targets for $d > 0$ and using the flow-matching loss (Equation 2) as a base case for $d = 0$. In principle, we can train the model on any

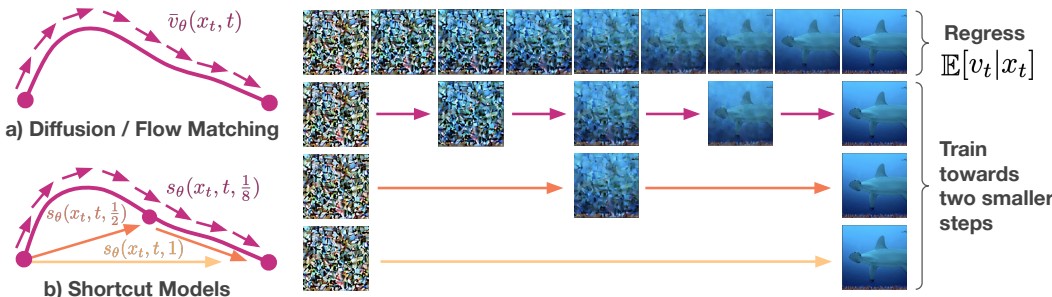

Figure 3: **Overview of shortcut model training.** At $d \approx 0$, the shortcut objective is equivalent to the flow-matching objective, and can be trained by regressing onto empirical $\mathbb{E}[v_t|x_t]$ samples. Targets for larger $d$ shortcuts are constructed by concatenating a sequence of two $d/2$ shortcuts. Both objectives can be trained jointly; shortcut models do not require a two-stage procedure or discretization schedule.

distribution of $d \sim p(d)$. In practice, we split the batch into a fraction that is trained with $d = 0$ and another fraction with randomly sampled $d > 0$ targets. We thus arrive at the combined shortcut model loss function:

$$\mathcal{L}^{\mathrm{S}}(\theta) = E_{x_0 \sim \mathcal{N},\, x_1 \sim D,\, (t,d) \sim p(t,d)} \left[ \underbrace{\|s_\theta(x_t, t, 0) - (x_1 - x_0)\|^2}_{\text{Flow-Matching}} + \underbrace{\|s_\theta(x_t, t, 2d) - s_{\text{target}}\|^2}_{\text{Self-Consistency}} \right],$$

$$\text{where} \quad s_{\text{target}} = s_\theta(x_t, t, d)/2 + s_\theta(x'_{t+d}, t+d, d)/2 \quad \text{and} \quad x'_{t+d} = x_t + s_\theta(x_t, t, d)d.$$

Intuitively, the above objective learns a mapping from noise to data which is consistent when queried under any sequence of step sizes, including *directly in a single step*. The flow-matching portion of the objective grounds the shortcut model at small step size to match empirical velocity samples. This ensures that the shortcut model develops a base generation capability when queried with many steps, exactly as an equivalent flow-matching model does. In the self-consistency portion, appropriate targets for larger step-sizes are constructed by concatenating a sequence of two smaller shortcuts. This propagates the generation capability from multi-step to few-step to one-step. The combined objective can be trained jointly, using a single model and over a single end-to-end training run.

## 3.1 TRAINING DETAILS

We now present a simple framework for training shortcut models via the objective described above. At each stage, we opt for design decisions which encourage training stability and simplicity.

**Regressing onto empirical samples.** As $d \to 0$, the shortcut is equivalent to instantaneous flow. Thus, we can train the shortcut model at $d = 0$ using the loss given by Equation 2, i.e. by sampling random $(x_0, x_1)$ pairs and fitting the expectation over $v_t$. This term can be seen as grounding the small-step shortcuts to match the data denoising ODE. We find that sampling $t \sim U(0,1)$ uniformly is the simplest and works as well as any other sampling scheme.

**Enforcing self-consistency.** Given that the shortcut model is accurate at small step-size, our next goal is to ensure that the shortcut model maintains this behavior at larger step-size. We rely on self-generated bootstrap targets for this purpose. To limit compounding approximation error, it is desirable to limit the total length of the bootstrap paths. We therefore opt for a binary recursive formulation in which two shortcuts are used to construct a twice-as-large shortcut (Figure 3).

We must decide on a number of steps $M$ to represent the smallest unit of time for approximating the ODE; we use 128 in our experiments. This creates $\log_2(128) + 1 = 8$ possible shortcut lengths according to $d \in (1/128, 1/64 \ldots 1/2, 1)$. During each training step, we sample $x_t$, $t$, and a random $d < 1$, then take two sequential steps with the shortcut model. The concatenation of these two steps is then used as the target to train the model at $2d$.

Note that the second step is queried at $x'_{t+d}$ under the denoising ODE and not the empirical data pairing, i.e. it is constructed by adding the predicted first shortcut to $x_t$, and not by interpolating

---

**Algorithm 1** Shortcut Model Training

**while** not converged **do**
   $x_0 \sim \mathcal{N}(0, I), \; x_1 \sim D, \; (d, t) \sim p(d, t)$
   $x_t \leftarrow (1 - t)\, x_0 + t\, x_1$       Noise data point
   **for** first $k$ batch elements **do**
     $s_{\text{target}} \leftarrow x_1 - x_0$       Flow-matching target
     $d \quad\;\; \leftarrow 0$
   **for** other batch elements **do**
     $s_t \quad\;\; \leftarrow s_\theta(x_t, t, d)$       First small step
     $x_{t+d} \leftarrow x_t + s_t\, d$       Follow ODE
     $s_{t+d} \leftarrow s_\theta(x_{t+d}, t + d, d)$       Second small step
     $s_{\text{target}} \leftarrow \text{stopgrad}(s_t + s_{t+d})/2$       Self-consistency target
   $\theta \leftarrow \nabla_\theta \|s_\theta(x_t, t, 2d) - s_{\text{target}}\|^2$

---

**Algorithm 2** Sampling

$x \sim \mathcal{N}(0, I)$
$d \leftarrow 1/M$
$t \leftarrow 0$
**for** $n \in [0, \ldots, M - 1]$ **do**
   $x \leftarrow x + s_\theta(x, t, d)\, d$
   $t \leftarrow t + d$
**return** $x$

---

towards $x_1$ from the dataset. When $d$ is at the smallest value (e.g. $1/128$), we instead query the model at $d = 0$.

**Joint optimization.** Equation 5 consists of an empirical flow-matching objective and a self-consistency objective, which are jointly optimized during training. The variance of the empirical term is much higher, as it regresses onto random noise pairings with inherent uncertainty, whereas the self-consistency term uses deterministic bootstrap targets. We found it helpful to construct a batch with significantly more empirical targets than self-consistency targets.

The above behavior also gives us room for computational efficiency. Training requires less self-consistency targets than empirical targets, and self-consistency targets are also more expensive to generate (requiring two additional forward passes). We can therefore construct a training batch by combining a ratio of $1 - k$ empirical targets with $k$ self-consistency targets. We find $k = (1/4)$ to be reasonable. In this way, we can reduce the training cost of a shortcut model to be roughly only $\sim 16\%$ more than that of an equivalent diffusion model[2].

**Guidance.** Classifier-free guidance (CFG; Ho & Salimans, 2022) has proven to be an essential tool for diffusion models to reach high generation fidelity. CFG provides a linear approximation of a tradeoff between the class-conditional and -unconditional denoising ODE. We find that CFG helps at small step sizes but is error-prone at larger steps when linear approximation is not appropriate. We therefore use CFG when evaluating the shortcut model at $d = 0$ but forgo it elsewhere. A limitation of CFG in shortcut models is that the CFG scale must be specified before training.

**Exponential moving average weights.** Many recent diffusion models use an exponential moving average (EMA) over weight parameters to improve sample quality. EMA induces a smoothing effect on the generations, which is especially helpful in the in diffusion modelling since the objective has inherent variance. We find that similarly in shortcut models, variance from loss at the $d = 0$ level can result in large oscillations in the output at $d = 1$. Utilizing EMA parameters for generating self-consistency targets alleviates this issue.

**Weight decay.** We find that weight decay is crucial for enabling stability, especially early on in training. When the shortcut model is near initialization, the self-consistency targets it generates are largely noise. The model can latch on to these incoherent targets, resulting in artifacting and bad feature learning. We find that proper weight decay causes these issues to disappear, and enables us to bypass the need for discretization schedules or careful warmups.

**Discrete time sampling.** In practice, we can reduce the burden of the shortcut network by only training on relevant timesteps. During training, we first sample $d$, then sample $t$ only at the discrete points for which the shortcut model will be queried, i.e. multiples of $d$. We train the self-consistency objective only at these timesteps.

---

[2]Approximating a backward pass as twice the compute of a forward pass. Each shortcut update uses 1 (forward) + 2 (backward) + (1/4)*2 (self-consistency targets) compute units, vs. 3 units in a diffusion update.

## 4 RELATED WORK

**Distillation of diffusion models.** A number of prior works have explored the distillation of pretrained diffusion models into a one-step or few-step model (Luo, 2023). Knowledge distillation (Luhman & Luhman, 2021) and rectified flows (Liu et al., 2022) generate a synthetic dataset by fully simulating the denoising ODE. As full simulation is expensive, a number of methods have been proposed that utilize bootstrapping to warm-start the ODE simulation (Gu et al., 2023; Xie et al., 2024). Alternatives to L2 distance have been proposed for distillation targets, such as adversarial (Sauer et al., 2023) or distribution-matching (Yin et al., 2024b;a) objectives. Our work most closely relates to techniques using binary time-distillation (Salimans & Ho, 2022; Meng et al., 2023; Berthelot et al., 2023), which divides distillation into $\log_2(T)$ stages of increasing step-size, shortening the required bootstrap paths. Unlike these prior works, we focus on learning a one-step generative model *end-to-end*, without a separate pretraining and distillation phase. Our method is computationally cheaper than full simulation methods (e.g. rectified flows, knowledge distillation) and avoids the multiple teacher-student phases of progressive distillation methods.

**Consistency modelling.** Consistency models (Song et al., 2023) are a family of one-step generative models that learn to map from any partially-noised data point to the final data point. While such models can be used as a student for distillation (Luo et al., 2023; Geng et al., 2024), *consistency training* has also been proposed to train consistency models end-to-end from scratch (Song et al., 2023; Song & Dhariwal, 2023). Our work tackles the same problem setting, but approaches it differently – consistency training enforces consistency among *empirical* $x_t$ and $x_{t+d}$ samples, which accumulates irreducible bias at each discretization step due to ambiguity. We instead sample $x'_{t+d}$ from the *learned ODE*, avoiding this issue. Shortcut models require only $\log_2(T)$ bootstraps, whereas consistency models require $T$. Shortcut models naturally support many-step generation as a non-bootstrapped base case, whereas consistency models require bootstrapping in all cases. In addition, shortcut models are practically simpler, as many of the consistency model tricks – e.g. using a strict discretization schedule, using perceptual loss rather than L2 loss – can be bypassed entirely.

## 5 EXPERIMENTS

We present a series of experiments evaluating quality, scalability, and robustness of shortcut models. In terms of quality, shortcut models show few- and one-shot generation capability competetive with prior two-stage methods, and outperform alternative end-to-end methods. At the same time, shortcut models maintain the performance of base diffusion models on many-step generation. We show that under increasing model scale, generation capability continues to improve. Shortcut models learn an interpolatable latent noise space, and are robust to applications such as robotic control.

### 5.1 HOW DO SHORTCUT MODELS COMPARE TO PRIOR ONE-STEP GENERATION METHODS?

In this section, we carefully compare our approach to a number of prior methods, by training each objective from scratch with the same model architecture and codebase. We utilize the DiT-B diffusion transformer architecture Peebles & Xie (2023). We consider CelebAHQ-256 for unconditional generation, and Imagenet-256 for class-conditional generation. For all runs, we utilize the AdamW optimizer (Loshchilov, 2017) with a constant learning rate of $1 * 10^{-4}$, and weight decay of $0.1$. All runs use the latent space of the `sd-vae-ft-mse` autoencoder (Rombach et al., 2022).

**Comparison to prior work.** We compare against two categories of diffusion-based prior methods: *two-stage* methods, which pretrain a diffusion model then distill it separately, and *end-to-end* methods, which train a one-step model from scratch over a single training run.

- **Diffusion (DiT-B)** represents a standard diffusion model, faithfully following the setup in (Peebles & Xie, 2023).

- **Flow Matching** replaces the diffusion objective with an optimal-transport objective, following Liu et al. (2022). Together with diffusion, these two methods provide a baseline for the performance of an iterative many-step denoising model. The rest of the methods all adopt the flow-matching objective as a base, using a standard flow-matching model as a teacher if applicable.

- **Reflow** provides a comparison to a standard two-stage distillation approach which fully evaluates a teacher model to generate synthetic $(x_0, x_1)$ pairs. We follow methodology from Liu et al. (2022), and generate 50k synthetic examples for CelebAHQ, and 1M synthetic examples for Imagenet. Each example requires 128 forward passes to generate. Reflow is comparable to knowledge distillation (Luhman & Luhman, 2021), except the student model is trained among all $t \in (0, 1)$ rather at just $t = 0$.

- **Progressive Distillation** provides comparison to a two-stage binary time-distillation approach, which aligns with our proposed method. Following Salimans & Ho (2022), starting from a pretrained teacher model, a series of student models are distilled, each with 2x larger step-size than the previous. To match our methodology, classifier-free guidance is used during the first distillation phase.

- **Consistency Distillation** compares to a consistency-model based two-stage distillation strategy. Pairs of $(x_t, x_{t+d})$ are generated via a teacher diffusion model, and a separate student consistency model is trained to enforce consistent $x_1$ prediction among these pairs.

- **Consistency Training** provides a comparison to a prior *end-to-end* method which trains a one-step model from scratch, most closely matching our setting. A consistency model is trained over *empirical* $(x_t, x_{t+d})$ samples from the dataset. Following Song et al. (2023), the time discretization bins are scheduled to increase over the training run.

- **Live Reflow** is an additional end-to-end method we propose, in which a model is trained on both flow-matching and Reflow-distilled targets simultaneously. The model is conditioned on each target type separately. Distillation targets are self-generated every training step using full denoising, so this method is considerably expensive computationally. We include it for comparison purposes.

Together, these works span the space of prior two-stage distillation and end-to-end training methods. To ensure rigorous comparison, we run all prior work comparisons on the same codebase, using an equal or greater amount of total computation than our method.

**Evaluation.** We evaluate models on samples generated with 128, 4, and 1 diffusion step(s), using the standard Frechet Inception Distance (FID) metric. We report the FID-50k metric, as is standard in prior work. Following standard practice, FID is calculated with respect to statistics over the entire dataset, no compression is applied to the generated images, and images are resized to 299x299 with bilinear upscaling and clipped to $(-1, 1)$. We use the EMA model parameters during evaluation.

Table 1 highlights the abilities of shortcut models to retain accurate generation under few- and one-step sampling. With the exception of two-stage progressive distillation, shortcut models outperform all prior methods, without the need for multiple training stages. Note that progressive distillation models lose the ability for many-step sampling, whereas shortcut models retain this ability. Unsurprisingly, diffusion and flow-matching models display poor performance on 4- and 1-step generation. Interestingly, in our experiments shortcut models display a slightly better FID than a comparable flow-matching model. A blind hypothesis is that the self-consistency loss acts as a form of implicit regularization of the model, although we leave this investigation to future work. Further visual examples are provided in Appendix A.

## 5.2 WHAT IS THE BEHAVIOR OF SHORTCUT MODELS UNDER VARYING INFERENCE BUDGETS?

We now analyze the behavior of shortcut models when images are generated with varying step counts. While naïve flow-matching models approximate the denoising ODE with a tangent velocity, shortcut models are trained to approximately integrate the denoising ODE, providing more accurate behavior at low step counts. Figure 4 shows that, while greater numbers of steps are always helpful, the degradation at low step counts is much more pronounced in naïve flow-matching models than in shortcut models. In our experiments, we find that many-step performance is not sacrificed, and shortcut models maintain the performance of baseline flow models even at high step counts.

Few-step artifacts in flow models resemble blurriness and mode collapse (Figure 1, left). Shortcut models generally do not suffer from such issues, and globally match the equivalent many-step images generated from the same initial noise. Artifacts in shortcut models resemble errors in high-precision details. One-step shortcut models can therefore be used as a proxy: if a downstream user wishes to refine a one-step generated image, they may simply regenerate the image from the same initial noise, but with more generation steps.

|  | CelebAHQ-256 (unconditioned) | | | Imagenet-256 (class conditioned) | | |
|---|---|---|---|---|---|---|
|  | 128-Step | 4-Step | 1-Step | 128-Step | 4-Step | 1-Step |
| **Two phase training** | | | | | | |
| Progressive Distillation | (302.9) | (251.3) | **14.8** | (201.9) | (142.5) | **35.6** |
| Consistency Distillation | 59.5 | 39.6 | 38.2 | 132.8 | 98.01 | 136.5 |
| Reflow | **16.1** | **18.4** | 23.2 | **16.9** | **32.8** | 44.8 |
| **End-to-end (single training run)** | | | | | | |
| Diffusion | 23.0 | (123.4) | (132.2) | 39.7 | (464.5) | (467.2) |
| Flow Matching | 7.3 | (63.3) | (280.5) | 17.3 | (108.2) | (324.8) |
| Consistency Training | 53.7 | 19.0 | 33.2 | 42.8 | 43.0 | 69.7 |
| Live Reflow (ours) | **6.3** | 27.2 | 43.3 | 46.3 | 95.8 | 58.1 |
| Shortcut Models (ours) | 6.9 | **13.8** | **20.5** | 15.5 | 28.3 | **40.3** |

Table 1: **Comparison of training objectives under equivalent architecture (DiT-B) and compute.** FID-50k scores (lower is better) are shown over 128, 4, and 1-step denoising. Shortcut models provide high-quality samples under any inference budget, within a single training run. Compared to diffusion and flow-matching, shortcut models drastically reduce needed sampling steps. Compared to distillation approaches, short models simplify the training pipeline and provides flexibility to choose inference budget after training. Parentheses represent evaluation under conditions that the objective is not intended to support.

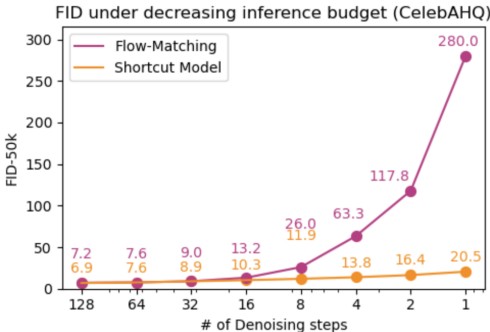

Figure 4: **Behavior of flow-matching and shortcut models over decreasing numbers of denoising steps.** While naïve flow-matching models incur degradation and mode collapse, shortcut models are able to maintain a similar sample distribution at few- and one-step generation. This capability comes at no expense to generation quality under large inference budget. See Figure 1 for qualatative examples.

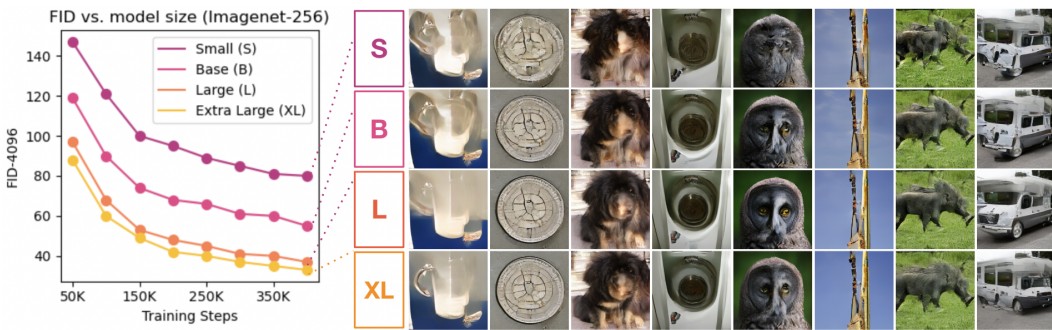

Figure 5: **One-step generation quality continues to improve as model parameter count increases.** While generative models tend to display continual improvement with model scale, bootstrap-based methods such as Q-learning have been shown to lose this property (Kumar et al., 2020). We show that shortcut models, while a bootstrap-based method, retains the ability to scale accuracy with model size.

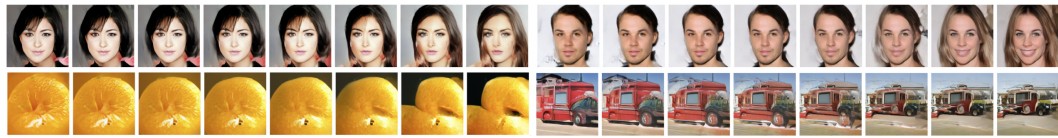

Figure 6: **Interpolations between two sampled noise points.** All displayed images are model generated. Each horizontal set represents images generated by one-step denoising of a variance-preserving interpolation between two Gaussian noise samples.

### 5.3 HOW DOES SHORTCUT MODEL PERFORMANCE INCREASE WITH MODEL SCALE?

A powerful trend in deep learning is that large over-parameterized models increase in capabilities when scaled up. This trend has been less clear in bootstrap-based methods such as learning Q-functions for reinforcement learning (Kumar et al., 2020; Sokar et al., 2023; Obando-Ceron et al., 2024). Common issues resemble a form of rank-collapse: by re-training on self-generated outputs, model expressivity may become limited. As shortcut models are inherently bootstrap-based, we examine whether shortcut models can extract a benefit from model scale. As highlighted in Figure 5, even in the one-step setting, our Transformer-based shortcut model architecture can achieve increasingly accurate generation as model size is increased. These results hint that even though shortcut models rely on bootstrapping, they nevertheless avoid drastic collapse and continue to scale with model parameter count. A shortcut model trained with DiT-XL size is able to achieve a one-step FID of **10.6** and a 128-step FID of **3.8** on Imagenet-256, as shown in Table 2 in the Appendix.

### 5.4 CAN SHORTCUT MODELS GIVE US AN INTERPOLATABLE LATENT SPACE?

Shortcut models represent a deterministic mapping from noise to data. To examine the structure of this mapping, we can interpolate between initial noise samples and view the corresponding generations. Figure 6 displays examples of interpolations generated in this fashion. A pair of noise points $(x_0^0, x_0^1)$ are sampled, then interpolated in a variance-preserving manner:

$$x_0^n = n x_0^0 + \sqrt{1 - n^2} x_0^1 \tag{6}$$

Despite no explicit regularization on the noise-to-data mapping, the resulting interpolated generations display a qualitatively smooth transition. Intermediate images appear to be semantically plausible. Note that in the visualized generations, all images are generated. While we do not explore interpolation between *existing* images, it may be possible to add noise to an existing image and interpolate those intermediate points, as is done by Ho et al. (2020).

### 5.5 DO SHORTCUT MODELS WORK IN NON-IMAGE DOMAINS?

The generative modeling objectives described in this work are all domain-agnostic, yet common benchmarks traditionally involve image generation. We now evaluate the generality of the shortcut models formulation by training shortcut model policies on robotic control tasks.

Specifically, we build off the methodology presented by Chi et al. (2023). This diffusion policy framework trains an observation-conditioned model to predict robot actions in an iterative manner. The original work uses 100 denoising steps, and we aim to reduce inference to one step by training a shortcut model instead. We utilize the same network structure and hyperparameters as the original work, except for changing AdamW weight decay from 0.001 to 0.1, and including a conditioning term for $d$. We additionally use a flow-matching target rather than an epsilon-prediction target.

Figure 7 highlights the robotic control tasks, Push-T and Transport, which were selected from Chi et al. (2023) as tasks where baseline methods struggle. We compare to iterative denoising methods IBC (Florence et al., 2022) and diffusion policy (Chi et al., 2023), as well as one-step methods LSTM-GMM (Mandlekar et al., 2021) and BET (Shafiullah et al., 2022). Results display the capability of shortcut model policies to achieve strong performance on robotic control, while limiting inference cost to a single function call. In contrast, one-step diffusion policies catastrophically fail.

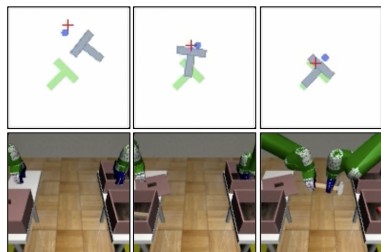

| | # Steps | Push-T | Transport |
|---|---|---|---|
| IBC | 100 | 0.90 | 0.00 |
| Diffusion Policy | 100 | **0.95** | **1.00** |
| LSTM-GMM | 1 | 0.67 | 0.76 |
| BET | 1 | 0.79 | 0.38 |
| Diffusion Policy | 1 | 0.12 | 0.00 |
| Shortcut Policy (ours) | 1 | **0.87** | **0.80** |

Figure 7: **Shortcut models can represent multimodal policies in a similar manner as diffusion policies, while reducing the number of denoising steps to 1.** Shown on the left are trajectories from the Push-T (top) and Transport (bottom) tasks. In each case, a generative model is trained on human demonstrations, and the model is queried to produce actions given a set of past observations.

## 6 DISCUSSION

This work introduces *shortcut models*, a new type of diffusion-based generative model that supports few-step and one-step generation. They key idea behind shortcut models is to learn a *step-size conditioned* model that is grounded to the data at a small step-size, and trained via self-bootstrapping at larger step-sizes.

Shortcut models provide a simple recipe that requires no scheduling and can be trained end-to-end in a single run. Unlike prior distillation methods, shortcut models do not require separate pretraining and distillation phases. In comparison to consistency training, shortcut models do not need a training schedule, use only standard L2 regression, and require less bootstrapping steps.

Empirical evaluations on CelebA-HQ and Imagenet-256 show that shortcut models outperform prior single-phase methods, and are competetive to two-stage distillation methods. Despite a reliance on bootstrapping, shortcut models are stable to train and scale with model parameter size. We showcase how shortcut models can be used in non-image domains, such as robotic control.

**Best practices.** For a practitioner wishing to utilize shortcut models, we recommend to keep the following in mind. Standard diffusion model practice should be followed – e.g., normalizing the dataset to unit variance. If the self-consistency loss displays erratic behavior, one can reduce the ratio of self-consistency targets in the training batch. A non-zero weight decay is highly recommended. Evaluating with EMA parameters provides a consistent improvement, but is not strictly neccessary.

**Implementation.** We provide a clean open-source implementation of shortcut models, along with model checkpoints, available at: https://github.com/kvfrans/shortcut-models

**Limitations.** While shortcut models provide a simple framework for training one-step generative models, a key inherent limitation is that the mapping between noise and data is entirely dependent on an expectation over the dataset. In other generative model flavors (e.g., GANs (Goodfellow et al., 2020), VAEs (Kingma, 2013)) it is possible to adjust this mapping, which has the potential to simply the learning problem. Secondly, in our shortcut model implementation there remains a gap between many-step generation quality and one-step generation quality.

**Future work.** Shortcut models open up a range of future research directions. In Table 1, shortcut models display slightly better many-step generation than a base flow model—is there a way for one-step generation to improve many-step generation (rather than typically the opposite)? Extensions to the shortcut model formulation may be possible that iteratively adjust the noise-to-data mapping (e.g., with a Reflow-like procedure (Liu et al., 2022)), or which reduce the gap between many-step and one-step generation. This work provides a stepping stone towards developing the *ideal* generative modelling objective—a simple recipe which satisfies the trifecta of fast sampling, mode coverage, and high-quality generations.

## 7 ACKNOWLEDGEMENTS

This work was supported in part by an National Science Foundation Fellowship for KF, under grant No. DGE 2146752. Any opinions, findings, and conclusions or recommendations expressed in this material are those of the author(s) and do not necessarily reflect the views of the NSF. PA holds concurrent appointments as a Professor at UC Berkeley and as an Amazon Scholar. This paper describes work performed at UC Berkeley and is not associated with Amazon. We thank Google TPU Research Cloud (TRC) for granting us access to TPUs for research.

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

# A   APPENDIX

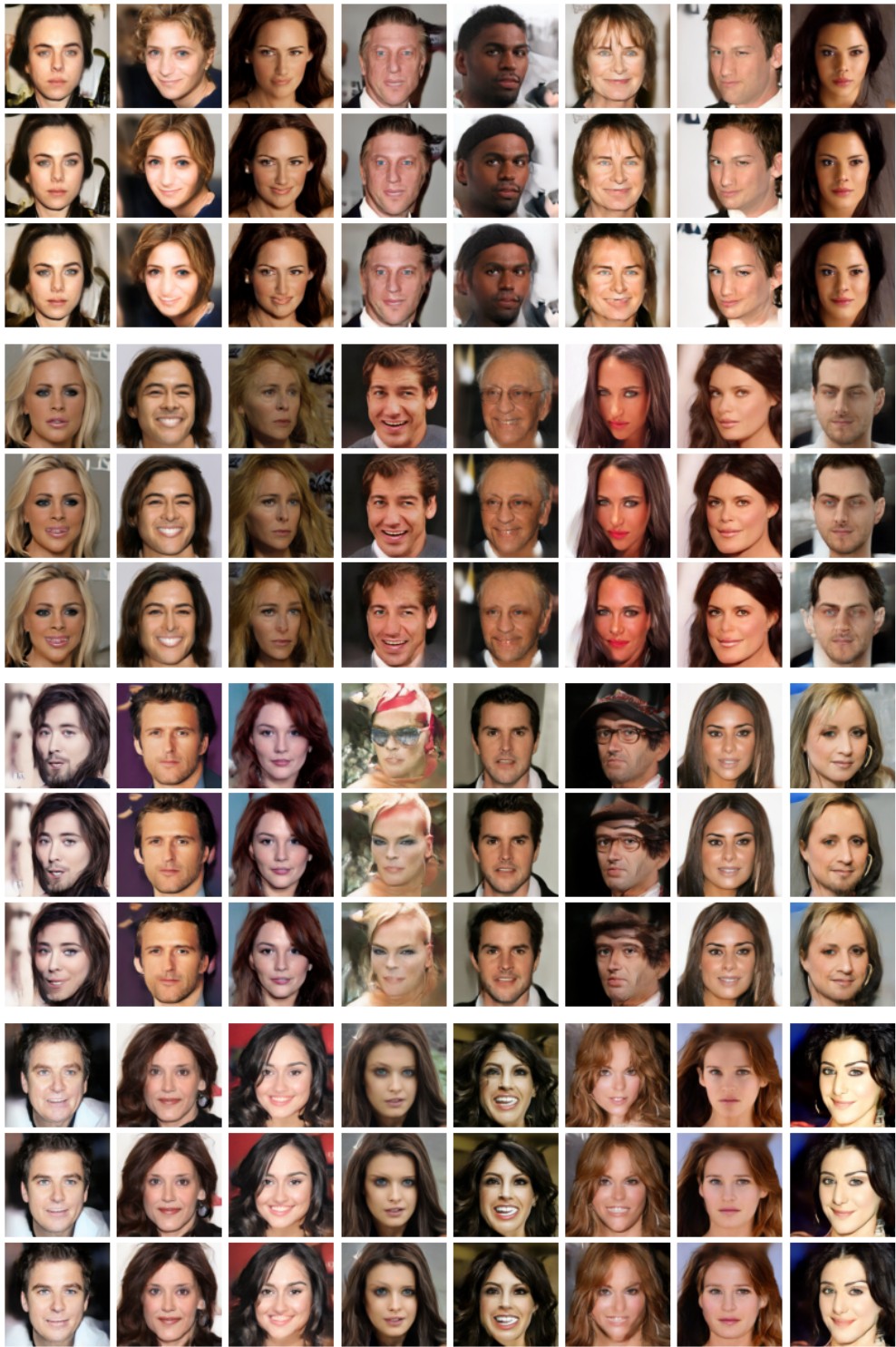

Figure 8: Examples generated from the unconditional CelebA-HQ dataset at 256x256 resolution. Each triplet represents an image generated from the same noise, at $(128, 4, 1)$ denoising steps from top to bottom. Results are generated from a DiT-B size model trained for 400k iterations.

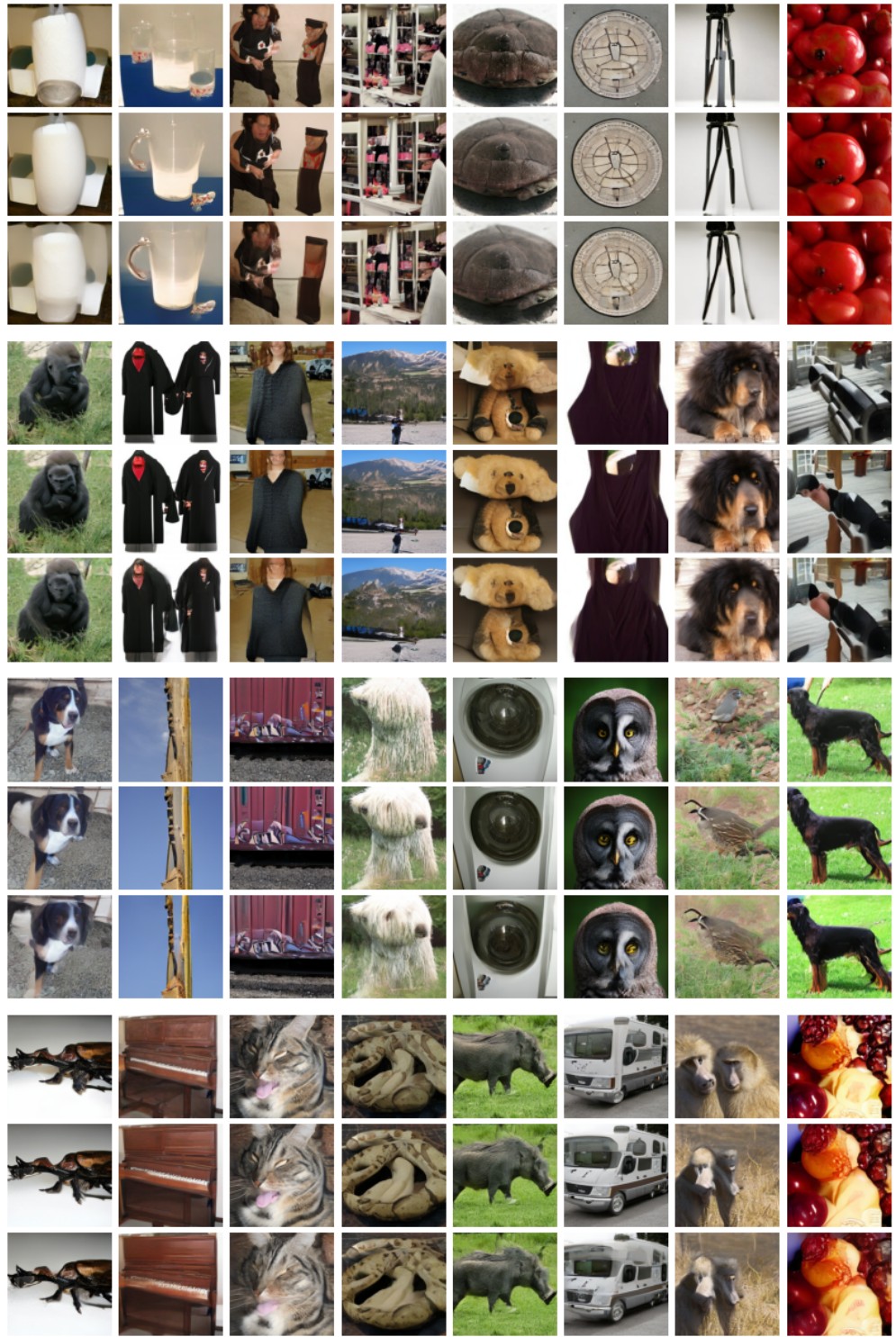

Figure 9: Examples generated from the class-conditional Imagenet dataset at 256x256 resolution. Each triplet represents an image generated from the same noise, at $(128, 4, 1)$ denoising steps from top to bottom. Results are generated from a DiT-XL size model trained for 800k iterations. CFG of 1.5 is used during the training process (not during generation).

| | FID | Sampling Steps | Param Count | Epochs Trained |
|---|---|---|---|---|
| Adversarial approaches | | | | |
| BigGAN-deep (Brock, 2018) | 6.96 | 1 | – | – |
| StyleGAN-XL (Sauer et al., 2022) | 2.3 | 1 | – | – |
| Denoising approaches | | | | |
| DiT-XL (Peebles & Xie, 2023) | 2.27 | 500 | 675M | 640 |
| ADM-G (Dhariwal & Nichol, 2021) | 4.59 | 250 | – | 426 |
| LDM-4-G (Rombach et al., 2022) | 3.6 | 500 | 400M | 106 |
| Shortcut Model (XL) | 3.8 | 128 | 676M | 250 |
| Shortcut Model (XL) | 7.8 | 4 | 676M | 250 |
| Shortcut Model (XL) | 10.6 | 1 | 676M | 250 |

Table 2: Comparison to state-of-the-art generative models on Imagenet-256. Due compute constraints, we cannot train models with the same compute as the best previously reported generative models. However, results demonstrate that **shortcut models are able to competitively scale up quality with model size**. The Shortcut Model (XL) using the DiT-XL architecture reaches substantially lower FID that the DiT-B architecture in Table 1, and is competetive with previous state-of-the-art models. Note that the compared models use different architectures and compute budgets. Moreover, parameter count is not completely proportional to training compute due to weight sharing, e.g. in convolutional and attention layers. Horizontal dashes indicate unreported quantities.

## B  TRAINING DETAILS

| | |
|---|---|
| Batch Size | 64 (CelebA-HQ), 256 (Imagenet) |
| Training Steps | 400,000 (CelebA-HQ) 800,000 (Imagenet) |
| Latent Encoder | sd-vae-mse-ft |
| Latent Downsampling | 8 (256x256x3 to 32x32x4) |
| Ratio of Empirical to Bootstrap Targets | 0.75 |
| Number of Total Denoising Steps (M) | 128 |
| Classifier Free Guidance | 0 (CelebA-HQ), 1.5 (Imagenet) |
| Class Dropout Probability | 0.1 |
| EMA Parameters Used For Bootstrap Targets? | Yes |
| EMA Parameters Used For Evaluation? | Yes |
| EMA Ratio | 0.999 |
| Optimizer | AdamW |
| Learning Rate | 0.0001 |
| Weight Decay | 0.1 |
| Hidden Size | 768 |
| Patch Size | 2 |
| Number of Layers | 12 |
| Attention Heads | 12 |
| MLP Hidden Size Ratio | 4 |

Table 3: Hyperparameters used during training. Model architecture follows that described in Peebles & Xie (2023), specifically DiT-B unless mentioned otherwise.

### B.1  COMPUTATION

All experiments are run on TPUv3 nodes, and methods are implemented in JAX. While runtimes vary per method, each training run typically takes 1-2 days to complete.

## B.2   METHOD DETAILS

Followed are a description of details behind the comparison methods. All methods are implemented in the same codebase, using the same architecture. Training budgets are chosen so that comparisons utilize roughly equal compute (or more compute if neccessary for the method to produce reasonable results).

**Flow-Matching.** We use the standard linear interpolation for generating velocity targets. We train the base model for 800k iterations, and denoise images using deterministic Euler sampling. A checkpoint at 400k iterations is used as a teacher model for two-stage distillation methods.

**Reflow.** We use a base flow-matching model to create the synthetic targets used in reflow. For CelebA-HQ, 50K targets are generated. 1M targets are generated for Imagenet. The distillation process is then run for 400k additional iterations on the synthetic dataset. CFG is used to generate synthetic data when applicable, and no CFG is used on the resulting distilled model.

**Progressive Distillation.** We run progressive distillation for the standard distillation time (400k iterations), split into equal sections for each phase of distillation. As there are 8 distinct phases, this results in 50k training steps per phase. At each phase, a teacher model is used to sample two sequential bootstrap steps, which a student model is trained to mimic. At the end of a training phase, the student becomes the new teacher. CFG is used only for the first phase (distilling 128-step model into a 64-step model).

**Consistency Distillation.** We train a consistency model in *velocity space*, to match the base framework of flow-matching. Thus at each iteration, consistency in velocity prediction is enforced between two points $(x_t, x_{t+d})$. The two points are computed by taking a small $d = (1/128)$ step from $x_t$ in the direction of the teacher flow model. Target velocities are computed from $x_{t+d}$ by querying the consistency model, using this prediction to estimate $x_1$, then computing target velocity as $(x_1 - x_t)/(1 - t)$.

**Consistency Training.** A consistency model is trained in the same format as described above. Instead of sampling $(x_t, x_{t+d})$ from a teacher model, the pairs are sampled by interpolating noise and empirical data samples at different strength. Following the methodology in Song et al. (2023), a discretization schedule is used. We use a binary-time schedule similar to that in progressive distillation. Specifically, the first phase has a single discretization interval, then two, then four, etc.

**Live Reflow.** In this method, we convert Reflow to a procedure that can be run in a single training run. Similar to a shortcut model, we train a $d$-conditioned model. A percentage of the batch will train the model at $d = 0$ towards the flow-matching loss. We use $0.75$ as this ratio, the same as the shortcut model. The other part of the batch consists of self-generated bootstrap targets generated by fully denoising a set of random noise. As this is a considerably expensive prodecure, we limit the number of denoising steps for target generation to 8. Even so, live reflow takes over 4x the computation of any other method.

