# OpenReview forum: "One Step Diffusion via Shortcut Models"
_ICLR.cc/2025/Conference — ICLR 2025 Oral_

### Official Review · Reviewer_NSZo · 2024-10-19

**Soundness:** 4
**Presentation:** 3
**Contribution:** 3
**Rating:** 8
**Confidence:** 4

**Summary:**

The paper introduces a shortcut design (self-consistency objective) in flow models for efficient and high-quality sampling. Unlike complex methods, this shortcut leverages existing networks and requires a single training phase.
The shortcut design involves incorporating an additional step size condition into the generative models, enabling them to learn the objective while considering the step size. This approach showcases improved performance in image and policy generation domains.

**Strengths:**

The paper introduces a shortcut design (self-consistency objective) in flow models to maintain consistency in intermediate sampling trajectories. This shortcut involves adding a self-consistency objective during training without significant additional burden. Empirically, it consistently outperforms other end-to-end methods in terms of generation quality.

**Weaknesses:**

Limited comparison with one-step generator GAN is provided. In Appendix Table 2, StyleGAN-X demonstrates the best performance (2.3). Does this suggest that GAN is currently the optimal choice for one-step sampling?

**Questions:**

* In Equation 5, Flow-Matching and Self-Consistency are intended to correspond to different units of time. Flow-Matching is represented by d=0, whereas Self-Consistency is associated with d > 0. Does it affect the training data samples or procedures?

* In Algorithm1, why stopgrad is applied in self-consistency target? Will it disable the gradient update?

* Table1 suggests two-phase training can give better results than all end-to-end methods in one step sampling: progressive distillation in 1-step:14.8 and 35.6 for Celeb and ImageNet. Does the two-phase method has higher performance ceiling than end-to-end?

* In Figure 7, does increasing the sampling steps lead to a higher success rate for Short-cut Policy, similar to the diffusion policy? In policy planning, the primary focus is on achieving multi-mode and high precision performance.

---

> ### Author Response · Authors · 2024-11-18
>
> Thank you for your review. Answering some questions:
> - Indeed, the SOTA one-step sampling methods are GAN-based to our knowledge. That said, this does not control for model size / training compute, which may be larger for those models. GANs have been known to be tricky to train, and the image-generation field generally has adopted diffusion modelling for these reasons.
> - Re: Equation 5, a key contribution from our paper is that d=0 and d>0 can be reconciled into a single objective. See Algorithm 1, which fully explains the procedure. The dataset does not need to be changed -- any standard unsupervised dataset works.
> - Stopgrad is applied when generating the targets, but not for matching the targets . This is standard practice in self-bootstrapped methods (e.g. Q-learning, DINO, etc). The rationale is that the targets are the source of truth, and the model should be trained to match these targets (the targets shouldn’t be changed to match the model, which would be backwards, so a stopgrad is used).
> - Two-phase training has a higher ceiling (when a training budget is specified *ahead of time*), in the sense that it is always possible to create a learning schedule that somewhat increases performance. In our setting we want to create an end-to-end method that is 1) simpler, requiring a single model and training loop, and 2) can be trained indefinitely long.
> - Re: Figure 7, we provide some additional results -- a multi-step shortcut policy can achieve scores of **0.93** (Push-T) and **0.9** (Transport). This shows we can increase accuracy with additional steps. There is still a gap between diffusion policy performance -- we believe the main cause is that we adopt a flow-matching schedule in our paper, whereas the original numbers use a tuned Gaussian schedule for the diffusion policy, and further hyperparameter tuning is required to match performance.
>
> If these clarifications and additional results have satisfied your questions, would you consider raising your review score?

---

> > ### Comment · Reviewer_NSZo · 2024-11-25
> >
> > Thank you for your comments, I raised my score and I think this is a good paper.

---

### Official Review · Reviewer_PitZ · 2024-11-03

**Soundness:** 3
**Presentation:** 3
**Contribution:** 3
**Rating:** 8
**Confidence:** 4

**Summary:**

The paper proposes denoising generative model called shortcut model. These models are trained end-to-end and can generate samples in both single-step and few-step setting. The core idea is to condition the model on step size in addition to time step, which is typically done in conventional diffusion models. The hypothesis is that the additional conditioning on step size allows shortcut models to account for the future curvature. The model is trained via a loss objective that combines the optimal-transport flow matching loss with a self-consistency loss. The self-consistency loss is enforced by ensuring that the prediction of model at a given skip size matches the target obtained by two sequential smaller skips (for the same skip distance) with the shortcut model. The high computational cost as well as variance during training is dealt with by computing self-consistency loss on a small subset of samples in the batch. The proposed method has been tested on CelebAHQ-256 and ImageNet-256, and has good single-step image generation quality.

**Strengths:**

1. _Simplicity_: The proposed method is simple, easy to understand and simplifies some challenges previously encountered in other methods to obtain one-step or few-step models. For eg. It gets rid of two-stage training required for diffusion distillation as well as complex training scheduling required for consistency models. The loss objective is intuitive and builds upon flow matching loss and introduces an additional self-consistency condition.
2. _Writing_: The paper is well-written. The core method has been explained well. The paper talks about challenges that can be encountered while training shortcut models and ways to address these challenges.
3. _Computational efficiency_: The method seems computationally efficient as with an additional (marginal) cost of training diffusion models, shortcut models can be trained to generate samples in single-step as well as few steps. Further, even on larger datasets such as ImageNet-256 and CelebAHQ-256, these models can be trained in 1-2 days (though the exact number of TPUv3 nodes is unclear).
4. _Applications_: The authors demonstrate advantages of this method on both image-generation as well as on robotic control tasks showing ways to generate an iterative sequence of actions in a single denoising step per action.

**Weaknesses:**

Note: The template of this paper seems to be different and doesn’t have line numbers present in the standard template for papers under review.

1. The proposed method for shortcut sampling is general and should work for any gaussian probability paths ($x_t = \alpha_t x_0 + \sigma_t \epsilon$) however this hasn’t been explored or described in the paper. The paper considers only one choice of optimal transport probability path ($x_t = (1-t) x_0 + t \epsilon$). However, nothing constraints the method to be restricted to this particular choice. (One motivation for this choice of parametrization could be that OT paths are said to have less curvature but in this case, these are conditional OT paths and are not exactly straight lines paths between $x_1$ and $x_0$.) It is worth exploring ablations on how shortcut models perform on other parametrization of gaussian probability paths/sampling paths. In addition, the text should also be updated to have a more general parametrization.
2. The paper should also include quantitative performance on more commonly used datasets such as ImageNet-64 and CIFAR-10. While reporting results on larger datasets such as CelebAHQ-256 and ImageNet-256 is impressive, reporting metrics on these smaller datasets is also important. Most of the prior papers report metrics on these datasets which makes it easier to compare the performance of shortcut models to a larger number of previously proposed methods/models which are not currently reported in the paper (Note: I understand that it is impossible to replicate all the previously proposed methods due to the vastness of literature in the space).
3. Minor: I personally feel that images such as in Figure 1 are misleading as the original flow-matching objective was not specifically meant for single step generation, and the performance is expected to deteriorate because these models don’t learn an ODE integrator. There are some advantages of the original flow-matching models as well over shortcut models. These models are continuous time models which can be evaluated at any time between [0, 1] whereas shortcut models are discrete time and can only be evaluated at specific discrete time steps.

**Questions:**

1. For the results of Progressive Distillation in Table 1, a reasonable 128-step model would be the obtained after taking the initial (say, T=1024) step model and then distilling it progressively until we get 128-step model. Was this model used to report metrics in the table or, did the authors use the final 1-step model to generate samples with 128 steps?
2. Appendix B.1 - could the authors specify the amount of TPUv3 nodes needed to run these experiments in 1-2 days?
3. For consistency training, is there a reason why the authors chose to deviate from the proposed discretization schedule by Song et al. (2023) during training? From my experience with these models, the choice of discretization schedule matters a lot, and deviations affect the performance. Further, does this paper implement consistency training (which includes perceptual loss)[1] or the follow up paper [2] which uses pseudo-Huber loss and proposes several improvements over their previous paper.
4. Could the authors include architectural details/positional encoding details of both of the conditioning (time/noise level and step size)?
5. Could the authors include some qualitative results of one-step generation of some other methods they have trained on in the appendix? FID score doesn’t always correlate with human perception of images.
6. Minor: Related work: for consistency models, the line needs revision - “shortcut models are practically simpler, as many consistency model tricks - e.g. using a strict discretization schedule, using perceptual loss rather than L1 loss - can be bypassed entirely. ” In their latest paper, Improved Consistency Training [2], Song et al. already bypass the need of perceptual loss; they use only pseudo-Huber loss.
7. Minor: some typos: competetive -> competitive (Section 5, line 2), qualatative -> qualitative (Figure 4 caption)

[1] Song, Yang, et al. "Consistency models." arXiv preprint arXiv:2303.01469 (2023).
[2] Song, Yang, and Prafulla Dhariwal. "Improved techniques for training consistency models." arXiv preprint arXiv:2310.14189 (2023).

---

> ### Author Response · Authors · 2024-11-18
>
> Thank you for the greatly detailed review. Addressing some points:
> - The shortcut sampling method is indeed general, and could in principle be applied to arbitrary time schedules, e.g. the Gaussian probability paths in DDPM. We chose the OT paths for simplicity, to match a flow-matching baseline. We will include a description of how this is possible in the paper, but we believe using the OT-based derivation helps simplify the equations for a first-time reader.
> - The point about shortcut models only being evaluatable at discrete steps is fair, and correct. In our practical experience, it is generally sufficient for quality-compute tradeoffs to only using sample steps in powers of 2 (e.g. 1 step for quick generation, 32 for more detailed, 128 for full detail), and the in-between values are often unused.
> - To us, Figure 1 has a clear rationale -- it's true that flow-matching models are not designed for few-step generation, but practitioners often attempt to use them this way anyways. We believe it's important to highlight the precise problem that shortcut models addresses, which is in bridging this gap.
> - For the progressive distillation, it is the final 1-step model. The initial base model is itself 128 steps, so the FID number for a 128-step progressive distillation model is actually the exact same as in the Flow-Matching column. The practical weakness of progressive distillation is that a separate model must be kept for different sampling steps -- we opt to only keep a single model for each method.
> - We use a v3-32 machine to run the experiments, which generally achieves 5-10 iters/sec for DiT-B size with the batchsize of 256 used in the paper.
> - The main reason for deviating from the original consistency-training schedule is because we use a different underlying time schedule (optimal transport / flow-matching) vs Gaussian probability paths in the consistency models paper. We also want to bias towards simplicity in our re-implementation. We attempt to compare methods fairly by stripping away architectural details where we can. For example, shortcut models could also be trained via LPIPS loss to push on FID numbers, but we opt to train all methods using only a standard domain-agnostic L2 loss.
> - Architectural details are included in the attached code, and we will also add a further description in the appendix. Positional encoding uses 2D sinusoidal encoding applied to each patch. The timestep and step-size are conditioned using AdaLN following DiT (Peebles 2022 https://arxiv.org/abs/2212.09748).
> - Attached are some visualizations of the one-step generations with other methods, all on the DiT-B size architecture. We will revise the Appendix to include these.
>
> New visualizations figure: https://i.imgur.com/KgzvFCO.jpeg
>
> - Thank you for the call-outs on typos and consistency models, we will fix them in the future revision!
>
> This was a thorough review and quite helpful to improving the work. We will include these clarifications in the next version of the paper. If the additional details and qualitative results have satisfied your questions, would you consider raising the review score?

---

> ### Comment · Reviewer_PitZ · 2024-11-24
>
> Thank you for your comments and for answering my questions. Thanks for providing the qualitative results. I hope the authors also include these images in the appendix of the paper as these are useful.
>
> As a practitioner myself, I humbly disagree with this point "it's true that flow-matching models are not designed for few-step generation, but practitioners often attempt to use them this way anyways". I think everyone understands that flows require solving an ODE.
>
> However, my major questions have been addressed. I believe that this is an interesting work and should be accepted. I do hope that the authors will update the paper to include a paragraph on the more general formulation.

---

### Official Review · Reviewer_w2mF · 2024-11-04

**Soundness:** 4
**Presentation:** 4
**Contribution:** 3
**Rating:** 8
**Confidence:** 4

**Summary:**

This paper proposes shortcut modes, an end-to-end training method that could produce high-quality images in either single or multiple sampling steps. The shortcut models take an additional input of the desired step size, compared to diffusion or consistency models.

**Strengths:**

* The paper is well written in the context of reducing sampling steps of diffusion models.
* The proposed method is novel and clearly presented.
* The experiments are comprehensive and convincing.

**Weaknesses:**

* Lacks explicit comparison of training compute against other methods.
* The issue with CFG could hinder future practical use.

**Questions:**

* Why the CFG scale must be specified before training?
* Why use 128 as the number of steps? What about something as large as 1024? How would this design choice affect the quality?
* Why not consider a two-stage distillation training, so that a pretrained diffusion model could be converted to a shortcut model?

---

> ### Author Response · Authors · 2024-11-18
>
> Thank you for the receptive review.
>
> - Regarding training compute for prior methods, indeed, although we normalize for # of gradient steps, certain methods require additional steps to generate bootstrap data, etc. We emphasize that **we are generous in our comparisons to prior methods, allowing them to use more compute per gradient-step.** For example, shortcut models requires ~16% more compute than standard diffusion. Consistency models require ~50% more (2 forward passes + 1 backward), progressive distillation requires 66% more (3 forward pases + 1 backward), and LiveReflow requires almost 32x more compute (due to fully denoising images every update).
> - The point regarding CFG is fair and is a limitation of the method. The nature of shortcut models requires self-distillation, thus, a desired CFG must be specified before training, as the bootstrap targets will depend on the desired CFG used to query the model at d=0. In principle, one could train a model that additionally conditions on a desired CFG (ala Meng 2022 https://arxiv.org/abs/2210.03142), but we did not try this.
> - In our settings and experiments, 128 is large enough that further increasing the # of steps only negligibly impacts FID. Thus for computational ease, we opt to use 128 as the limit for many-step generation.
> - It is possible to use a pretrained diffusion model and further train it with the shortcut-modelling objective, although we do not explicitly try this. To do this, one would need to adjust the architecture so that the network conditions on (d,t) rather than just (t).

---

> > ### Comment · Reviewer_w2mF · 2024-11-25
> >
> > My questions have been addressed, and thanks for your comments.

---

### Official Review · Reviewer_JHdH · 2024-11-06

**Soundness:** 3
**Presentation:** 3
**Contribution:** 4
**Rating:** 8
**Confidence:** 4

**Summary:**

The paper introduces shortcut models, a modification to the diffusion approach to be conditioned on noise level and step size. The approach is able to train end-to-end flexible models that simultaneously allow many-steps and one-step sampling.

**Strengths:**

- The paper is well written and easy to follow

- The proposed idea is novel and promising, this work pioneers a new family of diffusion/flow based generative models.

- The method is experimentally evaluated in a controlled setting, an extensive comparison with alternative methods is present.

- An open source codebase to replicate all the experiments and pretrained models will be release to falicitate further research

**Weaknesses:**

- The performance of 1 step generated samples still lags behind multi-step methods; 1-step generated images presents some artifacts so the usability of the proposed solution in a pratical setting is limited to generate proxy images with 1 step and resample them with multiple steps

- end-to-end training to reach a few sample diffusion model is an interesting solution as it removes the burden of performing two separate training phases and leads to a flexible model. However the best results in 1 step regime are obtained by Progressive Distillation, thus demonstrating that the best option for a one step model is to distill it in a separate training phase.

- It may be possible that learning few step models in a joint way introduces instabilities to the learning process especially when using much more complex data distribution than ImageNet or CelebA dataset. There are no guarantees on how the proposed approach would scale to large data regime training.

**Questions:**

- Comparing the various approach using the same compute budget could not be the best way as different methods may require different compute budget to converge to their optimal performance, did you take this consideration into account? It would be helpful to detail how the compute budget for each method has been fixed, and if you expect different convergence rates for each method. Moreover a plot of performance versus compute would be a nice addition if it's feasible.

- How does this method relates in any way to the ICLM 24 paper " Optimizing DDPM Sampling with Shortcut Fine-Tuning", I think the paper should be discussed as very relevant to the proposed method.

- It would be nice to introduce a comparison on additional compute required by post training distillation methods compared to the proposed solution which just increase the required compute by 16% over a standard diffusion training. A table highlighting the total compute overdue required by each method should be added.

- The method presents a lot of hyperparameters to be tuned for training, as the selection of M (total number of sampling steps) and k, have you ablated how varying this affect the final results? do you have estimated the impact of choosing a bigger/smaller k? Can you show any ablation study varying some of these parameters, and provide generale guidelines for their selection to pratictioners?

---

> ### Author Response · Authors · 2024-11-18
>
> Thank you for the receptive review. Addressing the weaknesses, we agree that these are all fair points, and have been mentioned in the Discussion/Limitations section of the paper as well. There is still quite some room for improvement in the 1-step performance setting. The concern about scalability is fair, however, note that we do consider a benchmark near the top complexity of image datasets examined in academic settings (Imagenet-256).
>
> Addressing questions:
> - We provide here extended plots on the convergence of different methods. On the left are from-scratch methods (ours included), and on the right are two-step distillation methods, which start from a pretrained flow-matching network. Note that these are FID-4096 scores (not FID-50k as in the paper) for computational reasons. We use the **same number of gradient steps and same architecture** for all comparisons.
>
> Additional FID plot: https://i.imgur.com/DLc0xHo.png
>
> - Following up on “additional compute required by post training distillation methods”, indeed, although we normalize for # of gradient steps, certain methods require additional steps to generate bootstrap data, etc. **We are generous in our comparisons to prior methods, allowing them to use more compute per gradient step.** For example, shortcut models requires ~16% more compute than standard diffusion. Consistency models require ~50% more (2 forward passes + 1 backward), progressive distillation requires 66% more (3 forward pases + 1 backward), and LiveReflow requires almost 32x more compute (due to fully denoising images every update).
> - We aim to reduce hyperparameters as much as possible. M is not very important, increasing M does not drastically improve performance, and 128 appears reasonable for all settings we examined. Tuning K is more sensitive. With a higher K, the model will emphasize d=0 more often, and 1-step performance suffers. Smaller K alleviates this, but is computationally slower.
> - Thank you for the reference to Shortcut Fine-Tuning, and we will address the relation in the related work. However, there are a number of significant differences between their methodology and ours. Their approach involves learning a critic, similar to GANs, then optimzing against the critic policy-gradients style, and their final evaluation is under 10-step denoising.

---

> > ### Comment · Reviewer_JHdH · 2024-11-22
> > **Reviewer Response**
> >
> > Thanks a lot for your comment, it addressed most of my concerns. I think this is a good paper that should be accepted.

---

### Public Comment · ~Lei_xing6 · 2025-04-28
**About equation 5**

About equation 5
s_{target} = s_\theta(x_t,t, d)/2 + s_{\theta}(x'_{t+d}, t,d)/2, the second term may have typo. The correct one may be
s_{target} = s_\theta(x_t,t, d)/2 + s_{\theta}(x'_{t+d}, t+d,d)/2  Because the second term should be starting from the time t+d rather than t

---

> ### Public Comment · ~Lei_xing6 · 2025-04-29
> **it is a confirmed typo**
>
> I check the source code at https://github.com/kvfrans/shortcut-models and confirm equation 5 is just a typo in the paper.  The code is correct.
>
>         v_b1 = call_model_fn(x_t, t, dt_base_bootstrap, bst_labels, train=False)
>         v_b2 = call_model_fn(x_t2, t2, dt_base_bootstrap, bst_labels, train=False)
> Obviously, the code uses t and t2 to represent different time. This is inconsistent with the equation 5 in the paper, but this is the correct one. Bingo!

---

### Meta-Review · Area_Chair_LRT5 · 2024-12-21

**Metareview:**

This paper provides a method for diffusion model distillation to one or multiple steps in one single training phase. It presents a simple and intuitive idea of conditioning the network on the number of desired steps and managed to make it work. All reviewers are very positive on their ratings. I believe this work presents significant contribution to the area of efficient diffusion models and recommend accept with a spotlight.

**Additional Comments On Reviewer Discussion:**

Most concerns raised by the reviewers are on the experimental details. And I think the authors did a great job responding in the rebuttal period, which reviewers also acknowledged.

---

### Decision · Program_Chairs · 2025-01-22

Accept (Oral)